# Using LC-MS/MS to Determine Salivary Steroid Reference Intervals in a European Older Adult Population

**DOI:** 10.3390/metabo13020265

**Published:** 2023-02-13

**Authors:** Sarah Gregory, Scott G. Denham, Patricia Lee, Joanna P. Simpson, Natalie Z. M. Homer

**Affiliations:** 1Edinburgh Dementia Prevention, Centre for Clinical Brain Sciences, University of Edinburgh, Edinburgh EH4 2XU, UK; 2Mass Spectrometry Core, Edinburgh Clinical Research Facility, Queens Medical Research Institute, University of Edinburgh, Edinburgh EH16 4TJ, UK; 3BHF/Centre for Cardiovascular Sciences, Queen’s Medical Research Institute, University of Edinburgh, Edinburgh, EH16 4TJ, UK

**Keywords:** LC-MS/MS, saliva, steroid hormone, supported liquid extraction

## Abstract

A number of steroids, including glucocorticoids and sex hormones, have been associated with neurodegenerative and cardiovascular conditions common in aging populations. The application of liquid chromatography tandem mass spectrometry (LC-MS/MS) steroid analysis offers an opportunity to conduct simultaneous multiplex steroid analysis within a given sample. In this paper, we describe the application of an LC-MS/MS steroid analysis method for the assessment of reference ranges of steroids in human saliva samples (200 µL) collected from older adults (age 50 years and above) enrolled in a European cohort investigating the risk for Alzheimer’s dementia. Saliva samples were prepared using supported liquid extraction (SLE) along with a calibration curve and analysed using a Waters I-Class UPLC (Ultra Performance Liquid Chromatography) and a Sciex QTrap 6500+ mass spectrometer. Mass spectrometry parameters of steroids were optimised for each steroid and a method for the chromatographic separation of 19 steroids was developed. Lower limits of quantitation (LLOQs), linearity and other method criteria were assessed. In total, data from 125 participants (500 samples) were analysed and assessed for reference ranges (64 male, 61 female). A total of 19 steroids were detected in saliva within the range of the method. There were clear diurnal patterns in most of the steroid hormones detected. Sex differences were observed for androstenedione (A4), testosterone (T), cortisone (E) and aldosterone (Aldo). In the first sample of the day, dehydroepiandrosterone (DHEA) was significantly higher in healthy volunteers compared to those with Alzheimer’s disease biomarkers. This LC-MS/MS method is suitable for the analysis of 19 steroids in saliva in adults.

## 1. Introduction

Steroid hormones play a central role in many biological functions throughout the life course. As our global population is aging, understanding the role of steroid hormones in both healthy aging and age-related illnesses is critical. Glucocorticoids have been associated in numerous studies with neurodegenerative conditions such as Alzheimer’s disease (AD) [1], Parkinson’s disease [2], stroke [3] and cardiovascular conditions, such as diabetes, that are common in later life [4]. Sex steroid hormones, particularly the estrogens, decline significantly in menopause, and have been related to cognitive function in older age [5], with both positive and negative associations seen between estradiol or estrone and cognitive domains including executive function, attention and psychomotor performance. Lower testosterone levels have been associated with AD [6,7], hypertension [8] and ischaemic heart disease [8]. 

Typically these studies have used plasma, serum and cerebrospinal fluid samples (CSF) for analysis; however, there is increasing focus on salivary sampling methods [9]. Saliva offers a non-invasive sampling technique when compared to other collection methods. This in turn confers two clear benefits, firstly in avoiding the cortisol response to venepuncture which can artificially inflate cortisol levels [10] and secondly eliminating the need for samples to only be taken in clinics with trained sample collection experts. Combining the non-invasive nature of the sample with the opportunity of home collection, it becomes feasible to collect multiple samples in a day. Given the circadian rhythm of many of the steroid hormones [11,12], particularly glucocorticoids [13], the opportunity to collect multiple samples as opposed to a single time point has potential to increase knowledge of the role steroids play in aging and disease. Previous research has demonstrated strong correlations between plasma and salivary steroids including cortisol [14,15], testosterone [11] and aldosterone [16]. Unlike in steroids in plasma, which can be protein-bound or free, salivary steroids are not protein-bound. Indeed, the fact that salivary glucocorticoids are free has clear benefits over other sampling techniques as a more suitable measure of adrenocortical function [17,18,19]. When serum cortisol concentrations are low, salivary cortisone levels have been demonstrated to be both detectable and a suitable alternative for serum cortisol levels [20,21]. In diseases of endocrine disruption, such as congenital adrenal hyperplasia, application of the measurement of five salivary steroids have been identified as markers of treatment control [22], showcasing the benefits of multi-steroid profiling in saliva samples in clinical research. 

Much of the previous analytical work on saliva samples has used immunoassays, particularly in the field of AD research. To use glucocorticoids as an example, the interpretation of salivary cortisol measurement by immunoassays may be challenged due to the presence of 11β-hydroxysteroid dehydrogenase (11β-HSD) type 2 in salivary glands which catalyses the conversion of cortisol to cortisone, such that cortisone is at a higher concentration in saliva than cortisol [23]. Methods of liquid chromatography tandem mass spectrometry (LC-MS/MS) allows for measurement of both cortisol and cortisone to be determined simultaneously. By exploiting the selective capabilities of LC-MS/MS cortisol and cortisone, they can be analysed as part of a wider steroid hormone panel [24,25]. Furthermore, steroid hormones that are present, but in low concentrations in saliva, require the sensitivity of LC-MS/MS analysis. This includes steroids such as estrone (E1), estradiol (E2) and 16-hydroxyestradiol (E3) [26]; these steroids, which further decrease with age, are implicated in diseases in women [27,28] and have the potential to serve as important biomarkers in saliva. In an editorial dating back to 2013, Handelsman and Wartofsky clearly extoll the benefits of a mass spectrometry approach and echo call for this to be the accepted method for the analysis of sex steroids [29]. Indeed, LC-MS/MS has now become the gold standard approach in many clinical diagnostics laboratories [30]. 

In this study, we developed an LC-MS/MS method for the analysis of multiple steroids in saliva and determined reference ranges by total sample, sex and AD biomarker presence. The steroids included corticoids (cortisol, F; cortisone, E; corticosterone, B; 11-dehydrocorticosterone, A, aldosterone, Aldo), androgens (testosterone, T; androstenedione, A4; dehydroepiandrosterone, DHEA; 5α-dihydrotestosterone DHT), progesterones (17α-hydroxyprogestogens, 17αOH-P4; progesterone, P4; 17α-hydroxypregnenolone, 17αOH-Preg; Pregnenolone, Preg), and the estrogens (estradiol, E2; estrone, E1; 16-hydroxyestradiol, E3). See Figure 1 for an overview of the steroids detected in the method. LC-MS/MS conditions were optimised such that the cortisol and cortisone measurement would be as sensitive and reliable as possible.

## 2. Materials and Methods

### 2.1. Materials and Chemicals

Cortisol (F), cortisone (E), corticosterone (B), 11-deoxycortisol (S), 21-deoxycortisol (21-DF), 11-deoxycorticosterone (11-DOC), testosterone (T), androstenedione (A4), 5α-dihydrotestosterone (DHT), dehydroepiandrosterone (DHEA), progesterone (P4), 17α-hydroxyprogesterone (17αOH-P4), pregnenolone (Preg), 17α-hydroxypregnenolone (17αOH-Preg), aldosterone (Aldo), estrone (E1), estradiol (E2), and 16-hydroxyestradiol (E3) were produced by Cerilliant and purchased from Sigma-Aldrich (Merck). 11-dehydrocorticosterone (A), was purchased from Steraloids, UK. Isotopically labelled internal standards 2,3,4-^13^C_3_-Cortisol (^13^C_3_-F), 2,3,4-^13^C_3_-Cortisone (^13^C_3_-E), 2,3,4-^13^C_3_-Corticosterone (^13^C_3_-B), 2,2,4,5,5,21,21,21-^2^H_8_-21-Deoxycortisol (d8-21-DF), 2,2,4,6,6-^2^H_5_11-Deoxycortisol (d5-11S), 2,3,4-^13^C_3_-Testosterone (^13^C_3_-T), 2,3,4-^13^C_3_-Androstenedione (^13^C_3_-A4), 2,2,3,4,4-^2^H_5_^-^Dehydroepiandrosterone (d5-DHEA), 2,2,4,6,6,17a,21,21,21-^2^H_9_-Progesterone (d9-P4), d8-17α-Hydroxyprogesterone 2,2,4,6,6,21,21,21-^2^H_8_-17a-hydroxyprogesterone (d8-17OHP4), 20,21-^13^C_2_,-16,16-^2^H_2_-Pregnenolone (^13^C_2_,d2-Preg), 2,3,4-^13^C_3_-Estradiol (^13^C_3_-E2), 2,3,4-^13^C_3-_Estrone (^13^C_3_-E1), were purchased from Sigma-Aldrich/Cerilliant. d8-Aldosterone (2,2,4,6,6,17,21,21-^2^H_8_-Aldosterone (d8-Aldo)) and 2,3,4-^13^C_3_-16-hydroxyestradiol (^13^C_3_-E3) were purchased from Cambridge Isotopes Laboratories/CK Isotopes and 2,3,4-^13^C_3_-5α-Dihydrotestosterone (^13^C_3_-DHT) was purchased from Isosciences (Ambler, PA, USA). All standards had reported purity factors of over 99.9%. 

All standards and corresponding isotopically labelled internal standards (IS) were kept at −20 °C. Quality controls MassCheck^®^ Cortisol, Cortisone saliva controls (lyophilised) for level 1 (0353) and level 2 (0354) were purchased from Chromsystems Instruments & Chemicals GmbH (Munich, Germany). Isolute SLE+400 96-well plates for extraction were provided by Biotage (Uppsala, Sweden) and 2 mL deep well 96-well collection plates were supplied by Waters (Wilmslow, UK). Methanol (MeOH, HPLC grade and LC-MS grade), 2-propanol (HPLC grade and LC-MS grade), water (LC-MS grade) and acetonitrile (LC-MS grade) were supplied by VWR (England, UK). Water (HPLC grade) and formic acid (LC-MS) grade were supplied by Fisher Scientific UK Ltd. (Leicestershire, UK). Ammonium fluoride (99.99%) was provided by Sigma-Aldrich (Gillingham, UK). Sample extraction was carried out using an Extrahera liquid handling robot, supplied by Biotage (Uppsala, Sweden). A Waters I-Class UPLC (Waters, Wilmslow, UK) and QTrap 6500+ mass spectrometer from AB Sciex (Macclesfield, UK) were used for LC-MS/MS analysis. The column used was a Kinetex C18 (2.1 × 150 mm, 2.6 µm particle size) and a Krudkatcher inline filter both purchased from Phenomenex (Macclesfield, UK).

### 2.2. Liquid Chromatographic Conditions

A Waters I-Class UPLC was used for the liquid chromatography on a Kinetex C18 2.1 × 150 mm, 2 µm particle size. The column temperature was maintained at 50 °C and the autosampler temperature was maintained at 10 °C. Mobile phase A consisted of water and 0.05 mM ammonium fluoride, mobile phase B consisted of methanol and 0.05 mM ammonium fluoride. A gradient elution was conducted at a flow rate of 0.3 mL/min over 16 min, starting at 55% B for 2 min, rising to 100% B over 6 min, held for 2 min, then returning to 55% B over 0.1 min and equilibrating for 4.9 min. The solvent flow was diverted to waste from 0–2 min and 11–16 min. The injection volume was 20 μL and the total analytical run time per sample was 16 min. 

### 2.3. Mass Spectrometric Conditions

Mass spectrometry multiple reaction monitoring parameters for each steroid were determined by infusing 1 µg/mL solutions of each steroid in 50% methanol into the Turbospray ionisation source of the QTrap 6500+ at 2 μL/min in both positive and negative ion modes. The product ion scans were used to determine four mass transitions, using the tune function of Analyst^®^ software, and the mass spectrometry method was built using these parameters in positive and negative mode. Solutions of individual steroids were injected into the LC system in turn while developing the best chromatographic separation of isomers and isotopomers, and the two most dominant mass transitions were selected for each steroid and isotopically labelled internal standard. Product ions that included the stable isotopes were preferentially selected for inclusion in the final LC-MS/MS method, where possible. Initial product ion scans for each steroid and internal standard are shown in Appendix A. For androst-4-ene-3-one analogues, we consistently found the common *m*/*z* 97 product ion, which is well reported in the literature [31,32], and ensured the chromatographic separation of all steroids to ensure the specificity of the method. LC eluent was diverted to the mass spectrometer between 2 and 11 min, with the remaining diverted to waste. Steroid hormones were detected with a QTrap 6500+ mass spectrometer from AB Sciex (Warrington, UK) equipped with an electrospray ionisation (ESI) turbo V ion spray source operated in both positive and negative modes. Multiple reaction monitoring (MRM) was used for quantitation of the steroids, where a quantification transition and a qualitative (confirmation) transition was monitored for each steroid. Positive ion spray voltage was set to 5500 V and negative ion spray voltage was set to −4500 V, with the source temperature maintained at 600 °C. The method included a curtain gas (nitrogen (N_2_)) of 30 psi, collision gas (N_2_) medium, air ion source gas 1 and air ion source gas 2 of 40 psi and 60 psi, respectively. Mass transition parameters for steroids and isotopically labelled internal standards are presented in Table 1 (positive ion multiple reaction monitoring (MRM)) and Table 2 (negative ion MRM). A chromatographic profile of steroids included in the method is presented in Appendix A. 

### 2.4. Saliva Samples

Saliva samples were collected into Sarstedt Salivettes^®^ (51.1534.500) (Sarstedt, Germany) from older adults participating in the European Prevention of Alzheimer’s Dementia (EPAD) longitudinal cohort study. The multi-site, pan-European, EPAD cohort study recruited healthy volunteers representing a spectrum of risk for future AD dementia. Well described in detail elsewhere [33,34], participants completed a number of study assessments during the baseline study visit, including providing saliva samples. Follow up visits occurred at month 6 and years 1, 2 and 3. Detecting AD earlier in the disease course and identifying intervention opportunities is currently an important research focus, and understanding the role of steroid hormones throughout the AD process is critical. 

Four saliva samples were collected by participants into Salivettes^®^ on the day of a clinical study visit. Participants were advised to collect samples at 09:00, 11:00, 15:00 and 22:00 on the day of collection, with the first and second sample collection typically under staff supervision at the study site. Samples were stored in a domestic fridge until returned to site. Samples contained in the Salivettes^®^ were initially stored at −80 °C prior to transfer on dry ice to long-term storage at −20 °C. Salivettes^®^ were centrifuged at 4500 rpm for 2 min, then saliva was aliquoted into labelled vials prior to preparation for steroid extraction. The saliva samples were kept, stored and analysed in adherence to procedures reviewed by ethical review boards. As this was a multi-centre international cohort study, each country and site was required to obtain review and approval relevant to local governance procedures prior to study activity beginning. 

Participants self-reported their sex as male or female. Participant disease status was categorized by the presence or absence of AD biomarkers. The AD biomarkers of phosphorylated tau-181 (pTau_181_) and amyloid-beta 1-42 (Aβ_42_) were analysed in CSF and used the fully automated cobas Elecsys^®^ AD portfolio platform (Elecsys^®^ Total -Tau CSF (roche.com) (accessed on 20 December 2022) [35,36]. Participants were defined as having AD if they had Aβ_42_ < 1000 pg/mL and/or pTau_181_ > 27 pg/mL, using cut offs as defined and used in previous analysis using the EPAD and other similar AD cohort datasets such as the Swedish BioFINDER and ADNI studies [37,38]. 

### 2.5. Calibrators, ISs and QC Samples

A steroid mixture was prepared using methanol (DHEA, P4: 50 µL × 1 mg/mL; T, A4:20 µL × 1 mg/mL; 17αOH-P4: 10 µL × 1 mg/mL; A, B, S, DHT, E1, E2, E3:5 µL × 1 mg/mL; 21-DF, 11-DOC, Aldo:50 µL × 100 µg/mL). The steroid mixture was used to prepare a calibration curve with 14 points. A stock solution of each IS was prepared in 100% MeOH. From these, a working solution containing 16 IS was prepared at 100 µg/mL.

Fourteen calibration standards were prepared in the concentration ranges of 0.005 to 5.0 ng/mL (F, E, DHEA, P4), 0.002 to 2 ng/mL (T, A4), 0.001 to 1.0 ng/mL (17αOH-P4, Preg, 17αOH-Preg) and 0.0005 to 0.5 ng/mL (A, B, S, 21-DF, 11-DOC, DHT, Aldo, E1, E2, E3). Aliquots of the calibrator, standards and IS solutions were stored at −20 °C until use. 

### 2.6. Sample Preparation 

A total of 200 µL of each calibration standard, quality control and 200 µL of saliva sample was added to the appropriate well on the 96-well plate. Standards, samples and QCs were spiked with 20 µL of internal standard solution. Each plate was sealed using a VWR 96-well-plate-sealing film and shaken on a plate shaker for 5 min at 600 rpm to ensure sufficient mixing of the standards and IS. Extraction was completed using an Extrahera liquid handling robot (Biotage, Uppsala, Sweden) with calibration standards, QCs and samples diluted 1:1 (*v*/*v*) with 200 µL 0.1% formic acid. The contents of the wells were then transferred to corresponding positions on the SLE+ 400 plate and loaded onto the sorbent using positive pressure. The plate was then extracted using 600 µL of dichloromethane:isopropanol (98:2 *v*/*v*) with elution under positive pressure. The process was repeated twice more, resulting in 1.8 mL of extract from each well. Following extraction, each elution plate was dried under nitrogen using a Biotage SPE Dry 96 Dual Sample Concentrator with the gas temperature set to 40 °C. Samples were re-suspended in 100 µL of Water:Methanol (70:30 *v*/*v*), sealed using a Waters adhesive plate-sealing film and shaken for 10 min at 600 rpm to ensure fully redissolved. 

### 2.7. Linearity and LLOQ

Each calibration curve contained 14 points representing appropriate concentration ranges for each steroid, along with double blanks and a solvent blank. Samples were prepared, extracted and analysed alongside each plate (28 plates analysed on 28 different days). The calibration curves were assessed using linear regression of the peak area ratio of the analyte to its respective IS against the concentration of the calibration point, with a weighting factor of 1/x. The intra-assay and inter-assay lower limit of quantitation (LLOQ) for each steroidwas defined as the lowest concentration that can be quantified within ± 20% bias of accuracy and less than ± 20% precision expressed as coefficient of variation (CV%), using six plates (with a calibration curve R > 0.99) to calculate.

### 2.8. Statistical Analysis 

Analyst ^®^ version 1.7 (AB Sciex) was used for instrument control and data acquisition. Data analysis was completed using MultiQuant version 3.0 (AB Sciex). Confidence intervals and reference ranges were calculated using R programming language, with the bias-corrected percentile confidence interval with bootstrapping (n = 999) using the boot package. Reference intervals were created for each sample for the whole cohort, for men and women separately and for those with and without AD biomarkers. Comparisons of median concentrations for each steroid between men and women and healthy volunteers and those with and without AD biomarkers were carried out using the Wilcoxon test. 

## 3. Results

### 3.1. Linearity, Lower Limit of Quantitation, Precision and Accuracy 

Six plates were selected to assess the precision and accuracy of the method across multiple analytical runs. The mean R value of the calibration curves for all steroids was consistent across the six plates (mean R > 0.99 for all steroids) with the exception of pregnenolone (Preg) where only four plates had calibration curves (mean R > 0.99). Where a steroid on a particular plate had a calibration curve <0.99 or was unable to be calculated, no data from that steroid were used in the analysis. The lower limit of quantitation (LLOQ), precision (relative standard deviation (RSD)) and accuracy (relative mean error (RME)) metrics are presented in Table 3. 

### 3.2. Reference Intervals

A total of 125 participants (64 male (51.2%), mean age 66.90 years (standard deviation (SD): 6.38) were included in the study, each providing four samples to give a total of 500 samples. Sample collection times were well aligned to the pre-specified protocol times (see Table 4). 

All 19 steroids were detected in at least one sample at each collection time point. This ranged from being detected in most samples (cortisone (E), samples 1, n = 115/125 (92.0%)) to only exceptionally detected (21-deoxycortisol (21-DF), sample 2, n = 2/125 (1.6%)). The 95% confidence intervals are presented for all steroids at each time point by full cohort, sex and disease status; however, those with a small sample size may be less representative (see Appendix A).

Considering glucocorticoids, there are clear diurnal patterns evident across the four sampling time points for cortisone (E), cortisol (F), 11-dehydrocorticosterone (A) and 11-deoxycortisol (S), with the reference intervals demonstrating a reduction in concentration from sample one to sample four. These diurnal patterns were observed in both men and women, as well as in healthy volunteers and participants with AD biomarkers. There were insufficient numbers of samples where concentrations were detected for corticosterone (B) and 21-deoxycortisol (21-DF) to determine diurnal changes in this steroid. There was a diurnal pattern observed for testosterone (T) concentrations with a reduction across the day. This decline was clearer in men than women, and in healthy volunteers compared to those with AD biomarkers. In contrast, progesterone (P4) increased throughout the day for both men and women, with more fluctuating patterns when the group was split by disease biomarker status. Androstenedione (A4) initially decreased between samples one and two before rising again for samples three and four. There was no clear diurnal pattern seen for aldosterone (Aldo), dehydroepiandrosterone (DHEA), pregnenolone (Preg) or 17α-hydroxyprogesterone (17αOH-P4). 

There were insufficient samples with detectable concentrations to infer diurnal rhythm patterns for 11-deoxycorticosterone (11-DOC), 17α-hydroxypregnenolone (17αOH-Preg) and 5α-dihydrotestosterone (DHT). There was some detection of the estrogens (estrone (E1), estradiol (E2) and 16-hydroxyestradiol (E3)); however, the sample size in which these were detectable remains comparatively small. Estrogen concentrations were detected in 60 of the 125 participants, two of whom were taking exogenous E2 with E1 detected in sample one and three for one participant, and E2 detected in sample four for the other participant. The remaining 58 participants did not report taking estrogen-containing medications.

There were statistically significant differences between men and women for the following steroids: androstenedione (A4) time point 1 (~08:18) and time point 4 (~21:43) (men had a higher median concentration than women); testosterone (T) time points 1–4 (~08:18; 11:24; 15:28; 21:43) (men had higher median concentrations than women); cortisone (E) time point 3 (~15:28) (men had a higher median concentration than women); and aldosterone (Aldo) time point 3 (~15:28) (women had a higher median concentration than men). One steroid showed differences by disease status for one sample: dehydroepiandrosterone (DHEA) time point 1 (~08:18) (healthy volunteers had a higher median concentration than participants with AD biomarkers). See Appendix A for full details.

## 4. Discussion

The LC-MS/MS method described in this study was assessed and found to be both precise and accurate for the quantitation of 19 steroids in a volume of 200 μL human saliva samples. 

Glucocorticoids were generally well detected, with the exception of corticosterone (B). Cortisone (E), cortisol (F) and 11-dehydrocorticosterone (A) all had a clear diurnal pattern with 95% confidence interval ranges including the highest concentrations in the morning with a gradual decline through the sampling period. Similar diurnal patterns were seen when median values were compared by sex and disease biomarker status. These diurnal patterns first reported by Weitzman, E.D. et al. (1971) [13] have been well documented throughout the literature. Findings from this study further support the importance of being able to collect multiple samples within a day to study diurnal rather than single time point glucocorticoid concentrations in saliva. Alternatively, collecting samples at time points that can be related to known reference intervals may be a more cost-effective option.

Cortisone (E) concentrations were notably higher than cortisol (F) concentrations at each time point and in all subgroup analyses. As discussed in the introduction, cortisone is found in higher concentrations in the saliva compared to cortisol, due to the increased expression of the 11β-HSD2 enzyme in the salivary glands [23]. The detection of cortisone and separation in the method from cortisol is clearly critical when analysing glucocorticoid concentrations in saliva, and demonstrates the value of the LC-MS/MS method. 

Salivary cortisol reference ranges have previously been published from the CIRCORT dataset, combining data from 15 studies to give age and sex-stratified ranges across the lifespan [39]. Comparing the reference ranges, there is typically a higher range at all time points in the relevant age groups in the CIRCORT data analysis compared to this present study. All studies contributing data to the CIRCORT dataset utilized immunoassays for sample analysis, meaning the higher values reported in the reference ranges may reflect interference from cortisone (E) and other steroids present in saliva. The cortisone (E) reference range and median values were lower in our study compared to results from the CIRCORT study, although only 20% of the reference group fell in the age group recruited in our study, meaning that this is likely to not be directly comparable [40]. Testosterone (T) and androstenedione (A4) median concentrations and reference ranges were comparable to previously published data [41,42,43]. Concentrations of dehydroepiandrosterone (DHEA) are higher in this study compared to a previous study investigating the effect of age on concentrations, although reference ranges were not calculated and so only the mean value was used in that study to compare with calculated 95% confidence intervals [43]. Reference ranges for progesterone (P4) and 17α-hydroxyprogesterone (17αOH-P4) corresponded to previously published reference ranges for ages 18–74 (no details on how many participants were in the same age range as this study) [44]. We were not able to identify published reference ranges for the remaining steroids (corticosterone (B), 11-dehydrocorticosterone (A), aldosterone (Aldo), 21-deoxycortisol (21-DF), 11-deoxycorticosterone (11-DOC), pregnenolone (Preg), 17α-hydroxypregnenolone (17αOH-Preg), 5α-dihydrotestosterone (DHT), 11-deoxycortisol (S), estrone (E1), estradiol (E2), 16-hydroxyestradiol (E3)) within the age group of interest for this study, highlighting the novel value of this work. 

Diurnal variation in sample concentrations was also noted in testosterone (T), 17α-hydroxyprogesterone (17aOH-P4), dehydroepiandrosterone (DHEA), progesterone (P4) and androstenedione (A4). This circadian variation is well described in the literature for all four of these steroids, although studies do suggest a loss of this rhythm with increasing age [45,46,47]. 

Previous studies have investigated further cyclical patterns of steroid hormones, for example, in pre-menopausal women in relation to the menstrual cycle, finding consistent changes in progesterone and estradiol levels throughout the cycle over the course of a year [48]. The women included in the sample used in our study were post-menopausal, with comparatively low-level concentrations of progesterone, estrone, estradiol and estriol detected, and as such, this was not a consideration for this study but should be kept in mind for any future work including pre- or peri-menopausal women. Another important cycle to consider is the seasonality of hormones. Whilst the majority of evidence for the seasonality of hormones is from animal research [49,50], a growing number of studies have investigated this in humans, finding evidence for seasonal changes in dehydroepiandrosterone, cortisol, testosterone and estradiol [51,52,53,54]. Future work should investigate associations between seasonal changes in steroids and disease, with seasonal affective disorder and rheumatological conditions already being studied with regard to seasonal steroid variations [55,56]. 

There were few significant differences between the steroid hormone concentrations by sex. The most notable difference was seen for testosterone (T) where all four samples across the day were higher for men compared to women. This finding is in line with multiple research studies [41,57,58]. The lack of consistent sex differences in steroid hormones where we may expect to see them (e.g., dehydroepiandrosterone [47]) may be due to an overall decrease in concentrations with age. Interestingly there was only one difference in salivary steroid hormone concentrations by AD biomarker status (dehydroepiandrosterone). A systematic review and meta-analysis including 31 studies found no differences in dehydroepiandrosterone concentrations between healthy controls and those with AD [59]. This meta-analysis included participants with established AD dementia, which is later in the disease process than the participants included in this study who had early AD but no dementia, and this may explain the differences observed in our results. It is interesting that there were no other differences in median concentration values between the healthy control and participants with AD biomarkers for the remaining glucocorticoids or testosterone in previous research findings [1,6,7,60,61,62], although it should be noted that these studies have used blood and cerebrospinal fluid samples analysed by immunoassays rather than salivary samples analysed using LC-MS/MS, and as such, may not be directly comparable. The participants who provided samples for this project included those with AD biomarkers but without overt symptoms of dementia, whereas the vast majority of studies to date have investigated more established AD. It may be that at this early stage of the disease process, there is not yet dysfunction in the hypothalamic pituitary adrenal (HPA) axis meaning glucocorticoid concentration levels remain comparable to healthy volunteers. 

Further work is needed to investigate associations between the salivary steroid hormones and diseases of aging, as well as with other indicators of AD. 

In conclusion, we developed an LC-MS/MS method for the simultaneous quantitation of 19 steroids in saliva and determined reference ranges, across 125 participants, with a mean age of 66.90 years, representing 74 healthy controls and 51 participants with AD biomarkers. This method can be applied to saliva samples in both healthy and disease clinical research studies to further interrogate and refine salivary steroid hormone reference intervals across the lifespan. 

## Figures and Tables

**Figure 1 metabolites-13-00265-f001:**
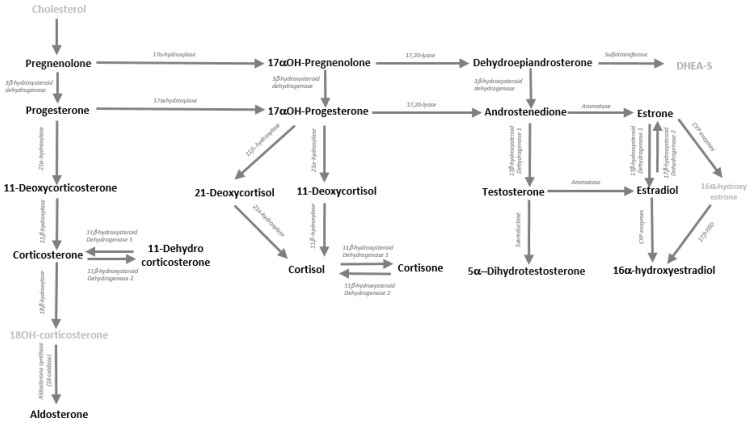
Human steroidogenesis pathway with steroids indicated in bold if measured in the human saliva samples by LC-MS/MS. Relevant enzymes indicated above the arrows. HSD is hydroxysteroid dehydrogenase.

**Table 1 metabolites-13-00265-t001:** Positive ion multiple reaction monitoring (MRM) parameters and retention times for each steroid and isotopically labelled internal standards, as analysed on a Kinetex C18 (150 × 2.1 mm; 2.6 μm) column on an Acquity I-Class UPLC and QTrap 6500+ mass spectrometer following electrospray ionisation. A4—Androstenedione; T—Testosterone; DHEA—Dehydroepiandrosterone; DHT—5α-dihydrotestosterone; P4—Progesterone; Preg—Pregnenolone; 17αOH-Preg—17α-hydroxypregnenolone; 17αOH-P4—17α-hydroxyprogesterone; 11-DOC—11-deoxycorticosterone; A—11-dehydrocorticosterone; S—11-deoxycortisol; 21-DF—21-deoxycortisol; B—corticosterone; E—cortisone; F—cortisol; DP—Declustering Potential; CE—Collision Energy; CXP—Collision Cell Exit Potential; RT-Retention Time. Quantifier (1) and Qualifier (2), MRM indicated accordingly.

Steroid	Internal Standard	Q1 Mass (*m*/*z*)	Q3 Mass (*m*/*z*)	DP (V)	CE (V)	CXP (V)	Time (Min)
A4 1	^13^C_3_-A4	287.1	97.0	61	27	14	6.9
A4 2		287.1	78.9	61	67	10	6.9
T 1	^13^C_3_-T	289.1	97.0	101	29	12	7.6
T 2		289.1	109.2	101	31	6	7.6
DHEA 1	d5-DHEA	289.1	253	121	15	46	8.1
DHEA 2		289.1	213.1	121	11	12	8.1
DHT 1	^13^C_3_-DHT	291.3	255.2	116	21	30	8.9
DHT 2		291.3	91.0	116	55	10	8.9
P4 1	d9-P4	315.0	97.1	96	23	10	8.9
P4 2		315.0	109.1	96	27	10	8.9
Preg 1	^13^C_2_,d2-Preg	317.1	281.1	66	31	12	10.3
Preg 2		317.1	159.0	66	29	12	10.3
17αOH-Preg 1	^13^C_2_,d2-Preg	333.1	297.1	36	13	22	9.6
17αOH-Preg 2		333.1	132.9	36	27	20	9.6
17αOH-P4 1	d8-17α-OHP4	331.1	109.0	66	29	12	8.1
17αOH-P4 2		339.1	100.1	66	31	12	8.1
11-DOC 1	d8-17α-OHP4	331.2	97.0	86	29	16	7.5
11-DOC 2		331.2	109.0	86	31	12	7.5
A 1	d4-F	345.1	121.0	66	31	12	3.6
A 2		345.1	91.2	66	83	40	3.6
S 1	d5-11S	347.1	97.0	71	27	12	5.7
S 2		347.1	109.0	71	33	16	5.7
21-DF 1	d8-21-DF	347.1	311.1	71	23	20	5.2
21-DF 2		347.1	269.0	71	27	14	5.2
B 1	^13^C_3_-B	347.1	121.1	76	29	8	5.3
B 2		347.1	90.9	76	75	12	5.3
E 1	d8-E	361.1	163.1	81	31	26	2.9
E 2		361.1	77.1	81	107	10	2.9
F 1	d4-F	363.1	121.2	76	31	8	3.5
F 2		363.1	91.1	76	83	10	3.5
**Internal Standards**							
^13^C_3_-Androstenedione		290.2	100.1	61	27	14	6.8
^13^C_3_-Testosterone		292.1	100.0	101	29	12	7.6
d5-Dehydroepiandrosterone		294.1	258.2	141	11	34	8.1
^13^C_3_-5α-DHT		294.2	258.3	116	21	30	8.9
^13^C_2_,d2-Pregnenolone		321.2	285.2	141	17	18	9.5
d9-P4		324.1	100.0	96	23	10	8.9
d8-17α-OHP4		339.2	96.9	66	29	12	7.9
d5-11-deoxycortisol		352.1	100.1	71	27	12	5.6
d8-21-deoxycortisol		355.2	319.1	71	23	20	5.1
d8-corticosterone		355.3	125.1	76	29	8	5.0
d4-cortisol		367.3	121.1	76	31	8	3.4
d8-cortisone		369.2	169.0	81	31	26	2.8

**Table 2 metabolites-13-00265-t002:** Negative ion multiple reaction monitoring (MRM) parameters for each steroid and isotopically labelled internal standards, as analysed on a QTrap 6500+ mass spectrometer following electrospray ionisation. E1—Estrone; E2—Estradiol; E3—16-hydroxyestradiol; Aldo—Aldosterone; DP—Declustering Potential; CE—Collision Energy; CXP—Collision Cell Exit Potential; RT—Retention Time. Quantifier (1) and Qualifier (2), indicated accordingly.

Steroid	Internal Standard	Q1 Mass (*m/z*)	Q3 Mass (*m/z*)	DP (V)	CE (V)	CXP (V)	Time (Min)
E1 1	^13^C_3_-Estrone	269.1	144.9	−150	−48	−15	7.2
E1 2		269.1	142.9	−150	−70	−15	7.2
E2 1	^13^C_3_-Estradiol	271.0	144.9	−110	−52	−21	7.0
E2 2		271.0	182.9	−110	−52	−19	7.0
E3 1	^13^C_3_-16OH-E3	287.1	171.0	−155	−48	−29	2.5
E3 2		287.1	145.0	−155	−54	−9	2.5
Aldo 1	d8-Aldo	359.1	188.9	−70	−24	−21	2.6
Aldo 2		359.1	331.0	−70	−22	−35	2.6
**Internal Standards**							
^13^C_3_-Estrone		272.0	147.8	−150	−48	−15	7.2
^13^C_3_-Estradiol		273.9	147.9	−110	−52	−21	7.0
^13^C_3_-16OH-Estradiol		290.2	173.9	−155	−48	−29	2.5
d8-Aldosterone		367.2	193.9	−70	−24	−21	2.6

**Table 3 metabolites-13-00265-t003:** Inter-assay validation of the lower limit of quantitation (LLOQ) of steroid hormones detected (n = 6). All data presented in A4—Androstenedione; T—Testosterone; DHEA—Dehydroepiandrosterone; DHT—5α-dihydrotestosterone; P4—Progesterone; Preg—Pregnenolone; 17αOH-Preg—17α-hydroxypregnenolone; 17αOH-P4—17α-hydroxyprogesterone; 11-DOC—11-deoxycorticosterone; A—11-dehydrocorticosterone; S—11-deoxycortisol; 21-DF—21-deoxycortisol; B—corticosterone; E—cortisone; F—cortisol; E1—Estrone; E2—Estradiol; E3—16-hydroxyestradiol; Aldo—Aldosterone; nM; nM = nanomolar, RME = relative mean error; RSD = relative standard deviation.

Steroid	LLOQ (ng/mL)	LLOQ (nM)	Intra-Assay %RSD	Intra-Assay%RME	Inter-Assay %RSD	Inter-Assay%RME
A4	0.500	1.75	3.5	13.2	3.4	3.1
T	0.050	0.17	5.7	12.3	4.7	1.4
DHEA	1.250	4.33	5.6	18.4	5.2	−2.5
DHT	0.125	0.43	10.5	4.8	9.7	6.2
P4	1.250	3.98	8.2	12.2	4.3	0.8
Preg	0.375	1.19	17.6	14.2	14.0	−3.1
17αOH-Preg	0.050	0.15	15.0	13.5	14.8	14.0
17αOH-P4	0.250	0.75	11.1	17.8	3.0	1.8
11-DOC	0.063	0.19	8.0	11.5	17.7	−13.7
A	0.125	0.38	7.6	16.5	6.5	6.2
S	0.125	0.36	5.4	10.3	8.4	4.6
21-DF	0.125	0.36	5.6	10.3	8.4	5.1
B	0.125	0.36	11.7	5.2	11.4	5.2
E	0.050	0.14	13.2	14.7	2.8	0.0
F	0.050	0.14	7.2	−14.4	8.6	−2.6
E1	0.063	0.23	4.8	10.2	19.2	−8.3
E2	0.125	0.46	4.9	9.3	8.6	2.8
E3	0.125	0.43	7.0	8.8	8.7	4.2
Aldo	0.063	0.17	4.0	10.5	18.4	−7.7

**Table 4 metabolites-13-00265-t004:** Mean sample collection times compared to protocol times and mean difference.

Sample Time Point Collection	Mean Collection Time hh:mm:ss (SD)	Protocol Time hh:mm:ss	Difference hh:mm:ss
Time Point 1	08:18:01 (0.05)	08:00:00	00:18:01
Time Point 2	11:24:52 (0.05)	11:00:00	00:24:52
Time Point 3	15:28:37 (0.05)	15:00:00	00:28:37
Time Point 4	21:43:53 (0.12)	22:00:00	00:16:07

## Data Availability

Data from the EPAD LCS are openly accessible and available via application at the following website: https://ep-ad.org/open-access-data/overview/. Data and samples used in the preparation of this article were obtained from the EPAD LCS dataset v.IMI (https://doi.org/10.34688/epadlcs_v.imi_20.10.30).

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
