# Peer review of "Using LC-MS/MS to Determine Salivary Steroid Reference Intervals in a European Older Adult Population"

_metabolites, 2023, doi:10.3390/metabo13020265_

Round 1

Reviewer 1 Report

Population aging is observed in highly developed countries all over the world. Due to the extension of life, the number of people suffering from neurodegenerative diseases is increasing. According to the estimates of the World Health Organization, up to 80 million people may suffer from dementia by 2030. Therefore, in an aging society, neurodegenerative diseases are and will be an even greater problem. Therefore, the reviewed manuscript is of great practical importance.

In the introduction, the authors made a thorough literature review in terms of the research problem under consideration. The purpose of the research was precisely defined. The authors simultaneously quantified 19 steroids in saliva using LC-MS/MS. Overall, the manuscript is interesting.

However, before publishing, authors must correct the manuscript:

1)     Is the LC-MS/MS method used for the quantification of the 19 steroids precise, reproducible accurate?

2)     Figure 1 is illegible.

3)     In chapter ‘2.1. Materials and chemicals', authors must state the purity degrees of the steroid standards used

4)     Table 5 takes up to 12 pages. I think it should be moved to Supplementary Materials.

5)     The authors at Supplementary Materials should provide sample chromatograms

Author Response

We would like to thank reviewer 1 for their time taken to review our manuscript and provide helpful comments. We have responded to these below and made relevant changes to the manuscript.  

  1. The LC-MS/MS method has been assessed for limits of quantitation prior to this study and inter-assay precision and accuracy calculated in a sub section of this study and results are presented. 
  1. Figure 1 has been updated with larger font size and  as a whole page diagram to increase readability, it will also be provided as a separate file to support with inclusion in the final manuscript if of higher visual quality. Small edits have been made to the figure to increase legibility of steroid name and enzyme name.  
  1. The steroid standards are certified reference materials and are purchased from external providers (Sigma/Cerilliant) and have certified concentrations, with purity factors over 99.9% for all certified reference materials. The manuscript has been updated with the following sentence (on Line 128): “All standards had reported purity factors of over 99.9%.”  
  1. Thank you for the suggestion to move Table 5 to supplementary materials, we have done so now. This is relabeled as Table S1.  
  1. A representative chromatographic profile of the steroids included in the method (representing the quantitative ions only) is included in the supplementary materials. Thank you for the suggestion 

Reviewer 2 Report

Dear the EditorGregory S et al reported an LC-MS/MS-based assay for steroid hormones in saliva. Overall, the assay procedure has been well documented and validated. These authors reported that the concentration of DHEA was higher in healthy individuals.Minor concerns:1) Fig. 1 appeared to be too small.2) In L134, is the concentrations of ammonium fluoride (0.05 mM) correct?

Author Response

We would like to thank reviewer 2 for their time taken to review our manuscript and provide helpful comments. We have responded to these below and made relevant changes to the manuscript.  

  1. Figure 1 has been updated to include as a whole page diagram to increase readability, it will also be provided as a separate file to support with inclusion in the final manuscript if of higher visual quality. Small edits have been made to the figure to increase legibility. 
  1. We can confirm this concentration of ammonium fluoride (0.05 mM) is correct for Mobile Phase A.  

Reviewer 3 Report

The article presents interesting data both on the detection and quantification of steroidal hormones in humans, in saliva, but also on the relationship of those hormones and neurodegenerative diseases, in elderly. However, as the authors stated, further work is needed to investigate associations between the salivary steroid hormones and diseases of aging.

It may be interesting that the authors present the MS/MS fragmentation spectra for the used standards, as a supplementary material. It would benefit the manuscript.

Tables should be formatted as the journal states. At the first submission, they were not formatted accordingly. The formatting of tables is not according to the template (font style, size).

Line 23 and elsewhere

“In total, data” there should be a comma.

Figure 1 should be definitely improved. There are some curvy underlinings, and also, the text is blurry and small.

Line 227

R programming language?

Table 5 and Table 6 should go to the supplementary materials.

Page 26, line 53

"(E3))" typo?

Page 25, line 15

“reported by [13]” mention who.

After minor modifications, the manuscript could be suitable for publication.

Author Response

We would like to thank reviewer 3 for their time taken to review our manuscript and provide helpful comments. We have responded to these below and made relevant changes to the manuscript.  

  1. We have now updated the text and included the requested additional product ion spectra and corresponding molecular structure of the steroid and isotopically labelled internal standards in the supplementary materials in two figures (Supplementary 1 and 2) for the positive ion product ions and the negative ion product ions, respectively. Text changes: Four product ions were initially selected for LC-MS/MS assessment under chromatographic conditions and the two transitions most consistent LC-MS/MS response and the with the least ion suppression were selected for the final method. For  androst-4-ene-3-one analogs we find the common m/z 97 product ion, well reported in the literature [31, 32] and ensure chromatographic separation of steroids to ensure specificity of the method. See product ion scans in Supplementary Materials for further detail. 
  1. Apologies for this, we had used the template but the tables did not seem to format accordingly, the updated manuscript is using the document version returned by the journal editors which uses the correct formatting  
  1. Line 23 (now line 24), comma added  
  1. Figure 1 has been updated to include as a whole page diagram to increase readability, it will also be provided as a separate file to support with inclusion in the final manuscript if of higher visual quality. Small edits have been made to the figure to increase legibility. 
  1. Line 227 (now line 531) has been updated to read: “…using R programming language, with…”  
  1. Thank for you the suggestion to move Tables 5 and 6 to supplementary materials, this has now been done. These are relabeled as Table S1 and Table S2.  
  1. Note (E3)) is not a typo (originally on Page 26, line 53, now on line 559), there is an open bracket on the previous page (line 556) which is closed here as well as the brackets for E3.  
  1. The typo originally on page 25, line 15 (now line 821-822) has been updated: “These diurnal patterns first reported by Weitzman, E.D., et al (1971) [13] have been well documented throughout the literature.” 

Round 2

Reviewer 1 Report

The authors revised the manuscript and therefore it can be published in its current form.